# SARS-CoV-2 Infection in Pregnancy: Clues and Proof of Adverse Outcomes

**DOI:** 10.3390/ijerph20032616

**Published:** 2023-02-01

**Authors:** Rosa Sessa, Simone Filardo, Luisa Masciullo, Marisa Di Pietro, Antonio Angeloni, Gabriella Brandolino, Roberto Brunelli, Rossella D’Alisa, Maria Federica Viscardi, Emanuela Anastasi, Maria Grazia Porpora

**Affiliations:** 1Microbiology Section, Department of Public Health and Infectious Diseases, “Sapienza” University of Rome, 00185 Rome, Italy; 2Department of Maternal and Child Health and Urology, “Sapienza” University of Rome, 00161 Rome, Italy; 3Department of Experimental Medicine, “Sapienza” University of Rome, 00161 Rome, Italy

**Keywords:** SARS-CoV-2, pregnancy outcomes, vertical transmission, neonatal outcomes

## Abstract

Background: Severe Acute Respiratory Syndrome Coronavirus 2 (SARS-CoV-2) represents one of the most threatening viral infections in the last decade. Amongst susceptible individuals, infected pregnant women might be predisposed to severe complications. Despite the extensive interest in SARS-CoV-2 research, the clinical course of maternal infection, the vertical transmission and the neonatal outcomes have not been completely understood yet. The aim of our study was to investigate the association between SARS-CoV-2 infection, obstetric outcomes and vertical transmission. Methods: A prospective observational study was performed, enrolling unvaccinated pregnant patients positive for SARS-CoV-2 (cases) and matched with uninfected pregnant women (controls). Maternal and neonatal nasopharyngeal swabs, maternal and cord blood, amniotic fluid and placenta tissue samples were collected; blood samples were tested for anti-S and anti-N antibodies, and histologic examination of placental tissues was performed. Results: The cases showed a significant association with the development of some obstetric complications, such as intrauterine growth restriction and pregnancy-associated hypothyroidism and diabetes, as compared to controls; their newborns were more likely to have a low birth weight and an arterial umbilical pH less than 7. The viral genome was detected in maternal and cord blood and placental samples in six cases. Conclusions: Pregnant women positive for SARS-CoV-2 infection are more likely to develop severe obstetric outcomes; their newborns could have a low birth weight and arterial pH. Vertical transmission seems a rare event, and further investigation is strongly needed.

## 1. Introduction

Severe Acute Respiratory Syndrome Coronavirus 2 (SARS-CoV-2) represents one of the most threatening viral infections in the last decade. At the beginning of 2020, the WHO declared a state of public emergency and, since then, the number of infected people has shown an exponential growth worldwide, with a total of about 625 million cumulative cases up to 26 October 2022 (https://covid19.who.int/, accessed on 26 October 2022). The overwhelming pressure on hospitalizations and the increased mortality rate have encouraged most countries to invest in SARS-CoV-2 research, providing a wide range of therapeutic options [1,2,3].

SARS-CoV-2 belongs to the Coronaviridae family and causes a respiratory infection characterized by a variety of different symptoms, such as fever, cough, fatigue, dyspnea, diarrhea and headache. Susceptible patients may develop severe lung injury, manifesting as bilateral interstitial pneumoniae or acute respiratory failure, often leading to fatal outcomes. These clinical conditions could be worsened by other comorbidities such as hypertension, obesity, diabetes, as well as pre-existing heart and respiratory diseases [4,5,6,7].

Amongst susceptible individuals, pregnant women might possess a higher risk of developing severe complications following SARS-CoV-2 infection. This is likely due to the several physiological adaptations occurring in pregnancy and involving the immunological, cardiovascular, respiratory and metabolic systems [4,8,9,10]. These can indeed result in a significant risk for fetal malformations, preeclampsia, fetal growth restriction, hypertensive disorders and diabetes [11,12,13,14,15,16,17]. Similarly to other coronavirus species, such as SARS and MERS, a correlation between SARS-CoV-2 and higher risk of miscarriage and stillbirth has also been reported [15,18,19,20,21].

SARS-CoV-2 infection is associated with a variety of clinical phenotypes, ranging from an asymptomatic course of disease to a massive interstitial pneumonia. Diagnostic imaging allows the clinician to detect the pulmonary involvement, usually via CT scan, although in pregnant women, lung ultrasound (LUS) represents an emerging method to assess pulmonary function, avoiding exposure to ionizing radiation [22,23].

Although the scientific literature has widely described the pathophysiological mechanisms of SARS-CoV-2 infection during pregnancy [24,25,26], the maternal consequences, the risk of vertical transmission and the related neonatal outcomes have not been completely investigated [27,28,29,30,31].

The aim of our study was to explore the association of SARS-CoV-2 infection, as well as the presence of related symptoms, with adverse obstetric outcomes in infected pregnant women. The potential SARS-CoV-2 vertical transmission was also evaluated.

## 2. Materials and Methods

### 2.1. Study Population

From December 2020 to December 2021, a prospective observational study was performed on pregnant women admitted to the Obstetrics ward of the Department of Maternal and Child Health of the “Policlinico Umberto I” Hospital, Sapienza University of Rome, Italy. The study design and protocol were approved by the ethical committee of the General Hospital Policlinico Umberto I, University of Rome “Sapienza” (Ref. 6275) and conducted according to the principles expressed in the Helsinki declaration of 1975 as revised in 2000. All patients provided written informed consent prior to sampling and provided a detailed personal, medical and gynecological history.

Inclusion criteria were:Age > 18 years,Unvaccinated pregnant women positive for SARS-CoV-2 infection prior or at the admission (Case group),Unvaccinated pregnant women without any previous or ongoing SARS-CoV-2 infection (Control group).Exclusion criteria were:Age < 18 years,Women positive for other infections during pregnancy, such as HIV, Hepatitis, Chickenpox and those related to the STORCH group (Syphilis, Toxoplasmosis, Rubella, Cytomegalovirus, Herpes Simplex virus, Varicella Zoster, Parvovirus B19).

The following characteristics were analyzed by reviewing patients’ medical records: maternal age, body mass index (BMI: weight/height^2^), gestational age, obstetric complications in pregnancy and during delivery.

At the admission, all the recruited patients underwent an obstetric ultrasound scan and maternal biological samples (blood, nasopharyngeal and vaginal swabs) were collected.

Vital signs, blood parameters, including white blood count, reactive c-protein, and procalcitonin; and maternal-fetal wellbeing were routinely checked in both groups. Moreover, the patients positive for SARS-CoV-2 infection at admission underwent lung ultrasound to evaluate the presence and severity of pneumonia, including the degree of ventilatory dysfunction and the severity of pulmonary status, in order to guide the following therapeutic strategies and improve pregnancy outcomes [22,23].

Asymptomatic SARS-CoV-2-positive patients were treated with enoxaparin at prophylactic dosage. All symptomatic cases and those with LUS score > 6 were, instead, treated with corticosteroids and enoxaparin at the therapeutic dosages.

When the SARS-CoV-2-positive women delivered, samples from cord blood, amniotic fluid (if available) and placental tissues were collected for SARS-CoV-2 RNA detection. The histologic examination of placental biopsies was also performed.

After delivery, neonatologists checked the newborns’ vital signs, umbilical artery pH, Apgar score and birth weight. A nasopharyngeal swab was also collected from all newborns.

### 2.2. Sample Storage and Analysis

At the admission, blood samples, as well as vaginal and nasopharyngeal swabs were sent to the Microbiology and to the Clinical Pathology laboratories for the detection of viral RNA and antibodies against SARS-CoV-2, respectively. After delivery, samples from cord blood, amniotic fluid and placental tissues were sent to the same laboratories.

Ten centimeters of placental tissue was sent to the Pathology department for histological examination to assess the presence of direct or indirect signs of viral infection and any other placental and/or amniochorial abnormalities.

All biological specimens were handled following the WHO’s guidelines for the containment of the SARS-CoV-2 pandemic. Specifically, vaginal swabs, blood and amniotic fluid samples were stored in sealed containers; vaginal swabs and amniotic fluid samples were directly stored at −80 °C. Conversely, blood samples were first centrifuged at 10,000× *g* for 15 min and then the serum was stored at −80 °C. Placental tissue specimens were directly stored in neutral buffered formalin solution and placed at room temperature (https://apps.who.int/iris/handle/10665/331058, accessed on 26 October 2022).

After birth, neonatal nasopharyngeal swabs were analyzed to detect the presence of viral RNA. Maternal and neonatal heath conditions were assessed daily, and blood samples were collected to check the inflammatory status and the development of complications during the hospitalization.

### 2.3. SARS-CoV-2 RNA Detection

Viral genome (RNA) was extracted from serum samples (200 μL), placental tissues (30 mg) and vaginal swabs via the MOLgen Universal Extraction Kit (Adaltis, Guidonia Montecelio, Italy), the RecoverALL Multi-sample Total Nucleic Acid Isolation Kit (ThermoFisher Scientific, Waltham, MA, USA), and the QIAmp Viral RNA Mini kit (Qiagen, Hilden, Germany), respectively, following the manufacturer’s instructions. In accordance with the WHO’s recommendations [32], the extracted RNA was then amplified via real-time Reverse Transcription Polymerase Chain Reaction (real-time RT-PCR) via the MOLgen SARS-CoV-2 Real Time RT-PCR Kit (Adaltis, Italy), following the manufacturer’s instruction. Three targets were identified: the N and E genes, specific for SARS-CoV-2, and the RNA-dependent RNA polymerase (RdRp) gene present in all sarbecoviridae. Real-time RT-PCR cycling conditions were 10 min at 45 °C for the Reverse Transcription, 2 min at 95 °C for Taq DNA-polymerase activation and 40 cycles of 15 s at 95 °C and 30 s at 60 °C for the DNA amplification. The CY5, ROX and FAM channels were used to detect the amplification of N, E and RdRp genes, and SARS-CoV-2 positivity was confirmed when either the N or E genes, alone or together to the RdRp gene, were detected at <38 CT. All samples were analyzed in triplicate.

### 2.4. Anti-SARS-CoV-2 Antibody Determination

Maternal and neonatal blood and amniotic fluid specimens were analyzed for the determination of antibodies against the SARS-CoV-2 virus. Each sample was tested using the ELISA method and evaluated as pan-antibodies; specifically, IgM, IgG and IgA targeting the nucleoprotein (N) (Elecsys anti SARS-CoV-2, Roche, Basel, Switzerland) and the receptor binding domain (RBD) in the S1 subunit of the Spike protein of the peak (pan-Ig anti-S1-RBD) (Elecsys anti SARS-CoV-2, Roche). The pan-antibodies Elecsys anti SARS-CoV-2 provides qualitative measurements with a sensitivity of 100.0% (95% CI 88.10–100.0) and a specificity of 99.81% (95% CI 99.65–99.91). A cut-off index (COI) ≥1 is considered as positive. Instead, the pan-Ig anti-S1-RBD Elecsys anti SARS-CoV-2 provides quantitative measurements with a sensitivity of 98.8% (95% CI 98.1–99.3) and a specificity of 99.98% (95% CI 9991%). The measurement range is 0.4–250 U/mL. A cut-off index ≥0.8 U/mL is considered as positive.

### 2.5. Histologic Examination of Placental Tissues

Placental tissues were analyzed via macroscopic and microscopic evaluation of the maternal and fetal interface. First, the placental disk dimensions (length, width and thickness), the characteristics of amniotic membranes, umbilical cord insertion and thickness were described. Then, the maternal vascular malperfusion, decidual arteriopathy, infarcts, fetal vascular malperfusion, chorionic vasculitis and thrombosis were assessed. The presence of chronic histiocytic intervillositis and villous trophoblast necrosis were also evaluated for their association with SARS-CoV-2 infection [29].

### 2.6. Statistical Analysis

Frequency data were presented as percentages, whereas continuous data as means and standard deviation. After assessing normal distribution via Shapiro–Wilk test, means were compared using a two-tailed student t-test for independent samples or a Mann–Whitney U test for non-parametric data. The Pearson chi-squared determined the association of frequencies among groups (Fisher’s exact test was used when any cell had expected values of <5), as well as the evaluation of odds ratio and 5% confidence intervals.

Statistical analysis was conducted using Excel software (Microsoft, version 2205) via the Real Statistics Resource Pack plugin (https://www.real-statistics.com/, version 8.2.1 accessed on 26 October 2022). Bonferroni correction was used when necessary. The single or multiple inference significance level was set at 5%.

## 3. Results

Two hundred four pregnant women were enrolled in the study. Among them, 68 were positive for SARS-CoV-2 via RT-PCR nasopharyngeal swab. One hundred thirty-six women were negative to SARS-CoV-2 infection, and they also did not report the viral infection during pregnancy. The main characteristics of the two groups are reported in Table 1 and Table 2.

Amongst the SARS-CoV-2-positive women (mean age 29.99 ± 5.3 years), 35.29% were symptomatic, whereas 64.71% were asymptomatic. All of them presented SARS-CoV-2 infection throughout the pregnancy (2.99% in the first trimester, 13.43% in the second and 83.58% in the third trimester). A total of 14.71% of them suffered from chronic diseases, and 44.12% also reported pregnancy-associated complications. Only 12.31% of cases delivered preterm. The mean weight of the neonates was 2982.27 ± 590.41 gr, and their APGAR scores were above 7 at 1 and 5 min in 93.85% and 100% of cases, respectively.

Amongst the SARS-CoV-2-negative women (mean age 32.82 ± 5.76 years), 17.91% suffered from chronic diseases and 18.38% reported pregnancy-associated complications. Only 9.56% of controls delivered preterm. The mean weight of their neonates was 3229.46 ± 457.84 gr, and their APGAR scores were above 7 at 1 and 5 min in 91.91% and 98.53% of cases, respectively.

A significant association of obstetric complications to the SARS-CoV-2-positive patients, as compared to the control group, was observed. In particular, SARS-CoV-2-positive patients seemed more likely to develop diabetes, hypothyroidism and intrauterine growth restriction during pregnancy than the controls (Figure 1). Furthermore, the neonates of mothers positive for SARS-CoV-2 were more likely to have a low birth weight (*p* = 0.004) and an arterial umbilical pH less than 7 (*p* < 0.00001) than controls.

The presence of symptoms seems to negatively affect the course of pregnancy and its outcome. In fact, among cases, symptomatic patients showed an increase in inflammatory indices, cesarean section rate and pregnancy-associated diseases compared to asymptomatic women, although no individual pregnancy-associated complications showed statistical significance (Figure 2).

In addition, statistically significant differences in the neonatal weight were found between symptomatic and asymptomatic patients. Indeed, babies from SARS-CoV-2-symptomatic mothers were more likely to have a lower birth weight than those from asymptomatic patients (2752.14 g vs. 3097.34 g, *p* = 0.022). Lastly, an arterial umbilical pH < 7 may be related to the presence of SARS-CoV-2 infection (*p* < 0.00001), rather than the onset of symptomatology (*p* = 0.87) (Figure 3 and Figure 4).

As for the vertical transmission of SARS-CoV-2, the viral genome was detected in three maternal (4.4%) and two umbilical cord (2.9%) blood samples, and one placental tissue specimen (1.5%). Conversely, the microbiological examination of vaginal swabs and amniotic fluid samples were negative for SARS-CoV-2 infection.

As for the histologic examination of placental biopsies, all symptomatic cases showed placental zones of vascular malperfusion and chronic intervillositis, as compared to controls.

Lastly, we did not observe any significant difference in the presence of anti-S and anti-N antibodies between symptomatic and asymptomatic cases. Nevertheless, an increasing trend in the detection of anti-N antibodies has been shown in women with symptoms, although with no statistical significance (Table 3).

## 4. Discussion

In the last few years, SARS-CoV-2 infection has been extensively studied, providing an initial knowledge on its pathophysiological mechanisms in pregnant women [24,25].

In this study, our attention focused on obstetric and neonatal outcomes of 68 unvaccinated pregnant women positive for SARS-CoV-2 infection, as compared to 136 controls. Our findings evidenced that patients affected by this viral infection are more likely to have serious complications and an adverse obstetric outcome, as is also confirmed by data in the literature [13,15,16,17,18,19,20]. In addition, an increasing rate of gestational diabetes was observed in our study, suggesting the potential viral entry into pancreatic beta cells via the angiotensin converting enzyme 2 (ACE2) receptors, leading, hence, to insulin imbalance and resistance, as previously described by Eberle et al. [33].

Further interesting data consists of the clear signs of malperfusion and chronic intervillositis observed in our placental tissue specimens, especially in pregnancies affected by a symptomatic SARS-CoV-2 infection. Indeed, SARS-CoV-2 infection has also been associated with the development of preeclampsia independently from the presence of symptoms, even though higher odds might involve women with a severe illness [34]. In this regard, malperfusion and chronic intervillositis might be an indirect sign of a huge inflammatory response that prevents physiological fetal growth [35], leading, for example, to low birth weight, as evidenced in the newborns of our symptomatic patients as compared to asymptomatic cases.

Concerning SARS-CoV-2 immune response, we observed an increase of anti-N antibodies in symptomatic patients, in accordance with the international literature. Indeed, Szymczak noticed that anti-N antibodies’ concentration is strongly associated with the severity of COVID-19 symptoms [36].

The tested newborns from SARS-CoV-2-positive mothers did not show any viral RNA in their nasopharyngeal swabs, hinting at the hypothesis of a potential involvement of different organs and tissue, rather than a preferential passage through the respiratory tract. This was also suggested by Kotlyar et al., who investigated neonates with a negative nasopharyngeal swab observing IgM in their blood [37].

As for SARS-CoV-2 vertical transmission, in our study the transplacental passage was found only in one patient with a monochorial diamniotic twin pregnancy complicated by a selective intrauterine growth restriction (sIUGR). To date, published studies on SARS-CoV-2 vertical transmission report contradictory results [26,38,39,40]. Several studies demonstrated the transplacental passage of SARS-CoV-2 viral particles through the maternal–fetal interface, with an estimated rate ranging from 3 to 8% of cases [37,41,42]. By contrast, other reports did not find any vertical transmission, evidenced by the absence of SARS-CoV-2 RNA in amniotic fluid, cord blood and pharyngeal swabs of the newborns [43,44].

Although SARS-CoV-2 vertical transmission seems to be a rare event, it cannot be excluded. As a matter of fact, the co-expression of ACE2 and TMPRSS2 was observed in multiple fetal tissues at different gestational ages during the second trimester, suggesting the potential viral entry into the bowel lumen through fetal swallowing of infected amniotic fluid [45]. Furthermore, the hypothesis of the pathogen transfer through the entire thickness of the maternal–fetal interface was recently suggested by the detection of SARS-CoV-2 virions in the syncytium-trophoblast and in the fetal capillary endothelium [39].

Overall, our data hint at the association between a symptomatic SARS-CoV-2 infection during pregnancy and adverse pregnancy outcomes, as well as low birth weight. In the future, it will be interesting to investigate the effect of SARS-CoV-2 vaccination on adverse obstetric outcomes, since SARS-CoV-2 vaccination was not recommended in Italy for pregnant women when we recruited the patients for this study.

## 5. Conclusions

Pregnant women affected by SARS-CoV-2 are more likely to develop severe obstetric outcomes and their newborns are more prone to have a low birth weight and low arterial pH, due to the clear signs of malperfusion and chronic intervillositis. Although SARS-CoV-2 vertical transmission seems to be a rare event, probably related to the potential viral entry into the bowel lumen through the fetal swallowing of infected amniotic fluid, it can still occur, although the pathophysiological aspects need further investigation.

## Figures and Tables

**Figure 1 ijerph-20-02616-f001:**
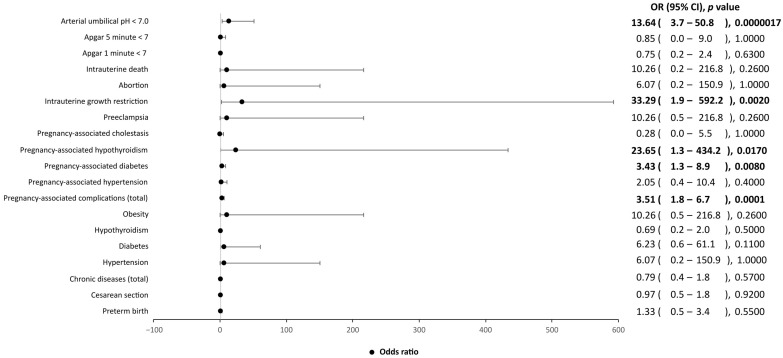
Influence of clinical factors, including chronic diseases, pregnancy-associated complications and Apgar scores, on SARS-CoV-2-positive mothers and their children, as compared to SARS-CoV-2-negative mothers and their children. OR, odds ratio; CI, confidence interval. Values in bold indicate statistical significance, as the level of statistical significance was set to *p* < 0.05.

**Figure 2 ijerph-20-02616-f002:**
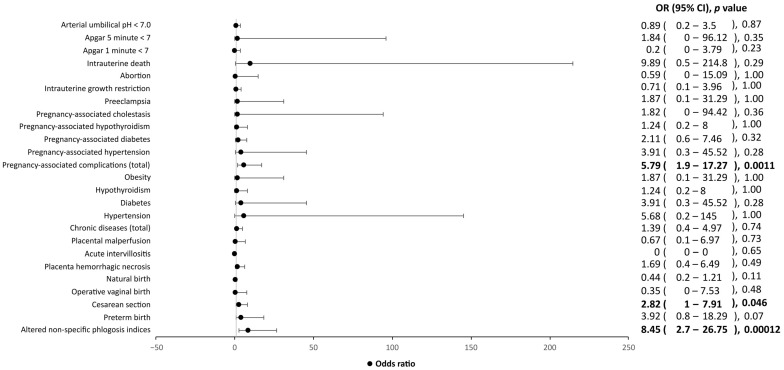
Influence of clinical factors, including chronic diseases, pregnancy-associated complications and Apgar score, on symptomatic SARS-CoV-2-positive mothers and their children, as compared to asymptomatic SARS-CoV-2 mothers and their children. OR, odds ratio; CI, confidence interval. Values in bold indicate statistical significance, as the level of statistical significance was set to *p* < 0.05.

**Figure 3 ijerph-20-02616-f003:**
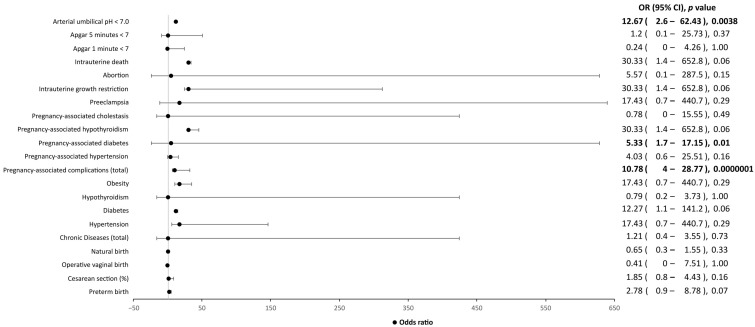
Influence of clinical factors, including chronic diseases, pregnancy-associated complications, and Apgar score, on symptomatic SARS-CoV-2-positive mothers and their children, as compared to SARS-CoV-2-negative mothers and their children. OR, odds ratio; CI, confidence interval. Values in bold indicate statistical significance, as the level of statistical significance was set to *p* < 0.05.

**Figure 4 ijerph-20-02616-f004:**
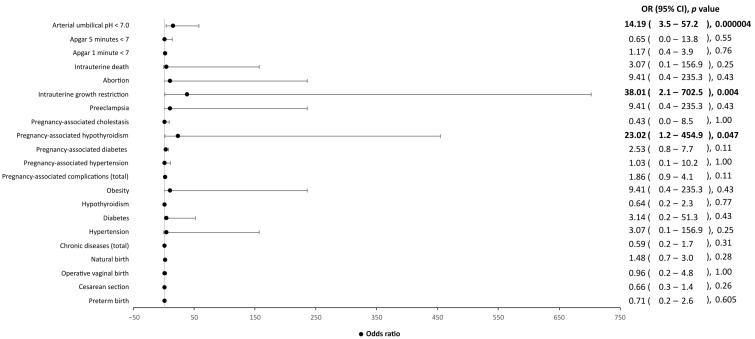
Influence of clinical factors, including chronic diseases, pregnancy-associated complications, and Apgar score, on asymptomatic SARS-CoV-2-positive mothers and their children, as compared to SARS-CoV-2-negative mothers and their children. OR, odds ratio; CI, confidence interval. Values in bold indicate statistical significance, as the level of statistical significance was set to *p* < 0.05.

**Table 1 ijerph-20-02616-t001:** General characteristics of SARS-CoV-2-positive pregnant women enrolled in the study and their children.

**SARS-CoV-2-positive women *(n* = 68)**	
*Age* (*average* ± *SD*)	29.99 ± 5.30
**Gestational age at first positivity (%)**	
*I trimester*	2.99
*II trimester*	13.43
*III trimester*	83.58
**Prevalence of chronic diseases (%)**	
*total*	14.71
*Hypertension*	1.47
*Diabetes*	4.41
*Hypothyroidism*	7.35
*Obesity*	2.94
**Prevalence of pregnancy-associated complications (%)**	
*total*	44.12
*Hypertension*	4.41
*Diabetes*	17.65
*Hypothyroidism*	7.35
*Cholestasis*	0.00
*Preeclampsia*	2.94
*Intrauterine growth restriction*	10.29
*Abortion*	1.47
*Intrauterine death*	2.94
**Non-specific phlogosis indices (%)**	
*Altered*	43.28
*Normal*	55.22
**Gestational length (%)**	
*Preterm*	12.31
*Full-term*	87.69
**Type of delivery (%)**	
*Cesarean section*	38.24
*Instrumental delivery*	2.94
*Vaginal delivery*	58.82
**SARS-CoV-2 manifestations (%)**	
*Symptomatic*	35.29
*Asymptomatic*	64.71
**SARS-CoV-2 antibody anti-N protein (%)**	
*>1 U/mL*	48.48
*<1 U/mL*	51.52
**SARS-CoV-2 antibody anti-S protein (%)**	
*>0.8 U/mL*	63.27
*<0.8 U/mL*	36.73
**Placenta hemorrhagic necrosis (%)**	
*Presence*	24.49
*Absence*	75.51
**Acute intervillositis (%)**	
*Presence*	8.16
*Absence*	91.84
**Placental malperfusion (%)**	
*Presence*	8.16
*Absence*	91.84
**Children from SARS-CoV-2-positive mothers *(n* = 66)**	
Weight (g, average ± SD)	2982.27 ± 590.41
**Apgar 1 min (%)**	
*<7*	6.15
*≥7*	93.85
**Apgar 5 min (%)**	
*<7*	0.00
*≥7*	100.00
**Arterial umbilical cord pH (%)**	
*Acidic (<7.0)*	23.53
*Basic (≥7.0)*	76.47
**SARS-CoV-2 antibody anti-N protein (%)**	
*>1 U/mL*	39.58
*<1 U/mL*	60.42
**SARS-CoV-2 antibody anti-S protein (%)**	
*>0.8 U/mL*	45.83
*<0.8 U/mL*	54.17

SD, Standard Deviation.

**Table 2 ijerph-20-02616-t002:** General characteristics of SARS-CoV-2-uninfected pregnant women enrolled in the study and their children.

**SARS-CoV-2 uninfected women (*n* = 136)**	
*Age* (*average* ± *SD*)	32.82 ± 5.76
**Prevalence of chronic diseases (%)**	
*total*	17.91
*Hypertension*	0.00
*Diabetes*	0.74
*Hypothyroidism*	10.29
*Obesity*	0.00
**Prevalence of pregnancy-associated complications (%)**	
*total*	18.38
*Hypertension*	2.21
*Diabetes*	5.88
*Hypothyroidism*	0.00
*Cholestasis*	2.21
*Preeclampsia*	0.00
*Intrauterine growth restriction*	0.00
*Abortion*	0.00
*Intrauterine death*	0.00
**Gestational length (%)**	
*Preterm*	9.56
*Full-term*	90.44
**Type of delivery (%)**	
*Cesarean section*	38.97
*Operative vaginal birth*	4.41
*Natural birth*	56.62
**Children from SARS-CoV-2 uninfected mothers (*n* = 136)**	
*Weight* (*g*, *average ± SD*)	3229.46 ± 457.84
**Apgar 1 min (%)**	
*<7*	8.09
*≥7*	91.91
**Apgar 5 min (%)**	
*<7*	1.47
*≥7*	98.53
**Arterial umbilical cord pH (%)**	
*Acidic* (<7.2)	2.21
*Basic* (≥7.2)	97.79

SD, Standard Deviation.

**Table 3 ijerph-20-02616-t003:** Presence of Anti-N and Anti-S antibodies towards SARS-CoV-2, associated with COVID-19 symptoms, in pregnant women and their children.

	SARS-CoV-2-Positive Pregnant Women (n = 49)						
	Symptomatic (*n* = 16)	Asymptomatic (*n* = 33)	*p*-Value	Odds Ratio	95% C.I.		
Antibody anti-N protein (%)	62.50	48.48	0.36	1.77	0.52	-	6.00
Antibody anti-S protein (%)	68.75	60.61	0.58	1.43	0.40	-	5.08
	**Children from SARS-CoV-2-Positive Mothers (n = 49)**						
	**Symptomatic (*n* = 15)**	**Asymptomatic (*n* = 33)**	***p*-Value**	**Odds Ratio**	**95% C.I.**		
Antibody anti-N protein (%)	33.33	42.42	0.75	0.68	0.19	-	2.43
Antibody anti-S protein (%)	33.33	51.52	0.24	0.47	0.13	-	1.68

C.I., Confidence Interval.

## Data Availability

The data presented in this study are available in Appendix A.

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
