# Peer review of "SARS-CoV-2 Infection in Pregnancy: Clues and Proof of Adverse Outcomes"

_ijerph, 2023, doi:10.3390/ijerph20032616_

Round 1
Reviewer 1 Report
This interesting study is a great addition to the scientific literature related to COVID-19’s impact on pregnancy. The following aspects are to be taken care of for improving your manuscript:
1) First of all, too many tables presented in the manuscript are monotonous to stimulate reading. The authors are advised to try to shorten the tables and/or re-organize some of the content in FIGURE format. An example of using figures like “Forest plot” can be found in the following reference:
Schwab R, Stewen K, Kottmann T, Theis S, Elger T, Hamoud BH, Schmidt MW, Anic K, Brenner W, Hasenburg A. Determinants of Pain-Induced Disability in German Women with Endometriosis during the COVID-19 Pandemic. Int J Environ Res Public Health. 2022 Jul 6;19(14):8277. doi: 10.3390/ijerph19148277. PMID: 35886130; PMCID: PMC9320034.
2) Apparently, COVID-19 vaccinated women with pregnancy as a control group were not included in the investigation. Can you add an explanation to the manuscript?
3) In L16 and L39, century should be changed to decade if that was what the authors meant.
4) SARS-CoV-2 is not used consistently throughout the manuscript, as can be seen in L144, L162, p.16(2nd line), and p.17 (last paragraph). Similarly, there are places where hyphens and space were used rather arbitrarily. For example, L24, p.14, and p.16 for anti-N, anti-S, and SARS-CoV-2.
5) Change table (L200) to Table.
6) On p.14 (1st paragraph), correct the decimal format used in 4,4% 2,9% 1,5%.
Author Response
First of all, too many tables presented in the manuscript are monotonous to stimulate reading. The authors are advised to try to shorten the tables and/or re-organize some of the content in FIGURE format. An example of using figures like “Forest plot” can be found in the following reference: Schwab R, Stewen K, Kottmann T, Theis S, Elger T, Hamoud BH, Schmidt MW, Anic K, Brenner W, Hasenburg A. Determinants of Pain-Induced Disability in German Women with Endometriosis during the COVID-19 Pandemic. Int J Environ Res Public Health. 2022 Jul 6;19(14):8277. doi: 10.3390/ijerph19148277. PMID: 35886130; PMCID: PMC9320034.
Thank you for your advice. We organized some of the results in Figures, as required. Moreover, we reduced the number of tables.
Apparently, COVID-19 vaccinated women with pregnancy as a control group were not included in the investigation. Can you add an explanation to the manuscript?
We designed the study in September 2020 and recruited patients between December 2020 and December 2021, after the approval of the Local Ethics Committee. COVID-19 vaccination was firstly introduced in Italy in January 2021 for healthcare professionals and from March 2021 it was progressively extended to the whole population. In Italy, the vaccination was recommended for pregnant women only in September 2021. For this reason, when we designed the study, we did not include vaccinated pregnant women as a control group. Therefore, we added in the text (page 19 line 390): “In the future, it will be interesting to investigate the effect of SARS-CoV-2 vaccination on adverse obstetric outcomes, since SARS-CoV-2 vaccination was not recommended in Italy for pregnant women when we recruited the patients for this study”
In L16 and L39, century should be changed to decade if that was what the authors meant
We changed “century” with “decade”.
SARS-CoV-2 is not used consistently throughout the manuscript, as can be seen in L144, L162, p.16(2nd line), and p.17 (last paragraph). Similarly, there are places where hyphens and space were used rather arbitrarily. For example, L24, p.14, and p.16 for anti-N, anti-S, and SARS-CoV-2.
We checked all the typing errors and standardized the spelling.
Change table (L200) to Table.
We changed “table” to “Table”.
On p.14 (1st paragraph), correct the decimal format used in 4,4% 2,9% 1,5%.
We changed the decimal format.
Reviewer 2 Report
Major Concerns:
-
English is atrocious. I’ve reviewed hundreds of papers but this is the first one I have ever actually seen a spelling mistake in the title. (should be clues and proof of adverse outcomes.) The paper needs proofreading by a native english speaker familiar with this area of medicine, or the authors need to consider publishing their work in an Italian language journal.
-
Scope is questionable. This is a paper related to infectious diseases and maternal fetal medicine (which is a subspecialty of Obstetrics and Gynecology.) This paper is not related to Environmental science IN THE LEAST, and is only loosely related to public health in the way that EVERYTHING IN MEDICINE is a little related to public health. In my opinion this should be transferred to a journal that has some form of patient care within its scope.
-
Paper contains many conclusions NOT supported by the data. The data shows statistically significant lower birth weights in babies born to mothers with covid. THATS IT. This is likely because of iatrogenic effects from doctors being afraid of keeping moms who have had covid pregnant. I’m not saying there is no value to this paper, but the conclusions consistently overstate that value.
Minor Concerns:
Line 28: (Multiple places after this.) Don’t have the “P Values” to suggest severe obstetrical outcomes are associated with COVID. The data does exist but not in this paper. You should only tell us what your data says.
Line 30: Cannot speak to the efficacy of vaccination in this paper. This line needs to be removed from both the conclusion in the abstract and the main conclusion of the paper.
Final Conclusion: The babies did not have “Inadequate weight,” they had lower birth weights. This was the only significant P value.
Author Response
English is atrocious. I’ve reviewed hundreds of papers but this is the first one I have ever actually seen a spelling mistake in the title. (should be clues and proof of adverse outcomes.) The paper needs proofreading by a native english speaker familiar with this area of medicine, or the authors need to consider publishing their work in an Italian language journal
As suggested by the Reviewer, we sent the manuscript to a native English speaker for revision. We fixed the title as recommended.
Scope is questionable. This is a paper related to infectious diseases and maternal fetal medicine (which is a subspecialty of Obstetrics and Gynecology.) This paper is not related to Environmental science IN THE LEAST, and is only loosely related to public health in the way that EVERYTHING IN MEDICINE is a little related to public health. In my opinion this should be transferred to a journal that has some form of patient care within its scope.
Although the Journal is titled “International Journal of Environmental Research and Public Health”, we sent the paper to a specific Special Issue titled “Improving Maternal and Child Health Outcomes". Furthermore, we have been invited to participate to this Special Issue.
Paper contains many conclusions NOT supported by the data. The data shows statistically significant lower birth weights in babies born to mothers with covid. THATS IT. This is likely because of iatrogenic effects from doctors being afraid of keeping moms who have had covid pregnant. I’m not saying there is no value to this paper, but the conclusions consistently overstate that value.
We are sorry we have not been clear, but not only the children birth weight, also the association of other clinical factors to SARS-CoV-2 positive mothers and their children resulted statistically significant, such as some pregnancy-associated complications, like hypothyroidism (p=0.017), diabetes (p=0.008) and intrauterine growth restriction (p=0.002), and arterial umbilical pH <7.0 (p=0.000001), as described in Results section. Nevertheless, we revised the discussion and conclusions in light of the considerations of the Reviewer.
Line 28: (Multiple places after this.) Don’t have the “P Values” to suggest severe obstetrical outcomes are associated with COVID. The data does exist but not in this paper. You should only tell us what your data says
As described in results section, we found that some pregnancy-associated complications, like hypothyroidism (p=0.017), diabetes (p=0.008) and intrauterine growth restriction (p=0.002), were significantly associated to SARS-CoV-2 pregnant women
Line 30: Cannot speak to the efficacy of vaccination in this paper. This line needs to be removed from both the conclusion in the abstract and the main conclusion of the paper.
We removed the statement “Spreading vaccination across the globe could limit the impact of this viral infection on pregnancy’s course, minimizing the possibility of short and long-term neonatal complications” from the Abstract and Conclusion sections
Final Conclusion: The babies did not have “Inadequate weight,” they had lower birth weights. This was the only significant P value.
We changed the expression “inadequate birth weight” in Conclusion section (see page 19, line 401).
Reviewer 3 Report
The reviewed article: „SARS-CoV-2 infection in pregnancy: clues and proofs of adverse outcomes” is related to the problem that has taken over the world - the SARS-CoV-2 pandemic. This publication presents the results of research related to: SARS-CoV-2 virus infection in pregnant women, vertical transmission of the virus from mother to neonate, complications related to the neonates born to SARS CoV-2 infected mothers. The authors of the article properly defined the research problem, which allowed to obtain reliable results and identified the association between SARS-CoV2 infection, obstetric outcomes and vertical transmission. The research methodology is accurate and the study group properly selected (large study and control group) according to the purpose of the study. Patients were included in the study in accordance with the protocol approved by ethical committee of the General Hospital 84 Policlinico Umberto I, University of Rome “Sapienza” (Ref. 6275). Inclusion and exclusion criteria are apropriately selected for the subject of the study - patients from both (the control and the study) group participating in the study signed consents and confirmed that they had not been vaccinated against SARS-CoV-2 infection. Performed examinations were carried out in accordance with the established diagnostic methods. Biological materials were properly stored and secured, samples taken to identify infection were collected and stored in appropriate conditions in accordance with WHO guidelines and evaluated by specialized diagnostic laboratories. Keeping in mind that SARS-CoV-2 is global-wide infection and the population of women participating in the study was identified, this research might encourage other research groups to expand the following subject by including populations in other parts of the world. The conclusions in the article are consistent with the results contained in the table (Table 5. Characteristics of symptomatic SARS-CoV-2 positive mothers and their children as compared to healthy controls) however, in order to properly assess their pathophysiology, each topic should be widened. Never the less this article might be an introduction to further challenges and scientific researches on the correlation of SARS-CoV-2 infection with complications in pregnant women and neonates. According to the paragraphs 4 (Discussion) and 5 (Conclusions) contained in the article and the conclusions detailed by the researchers, COVID-19 vaccines are recoemnded for pregnant women and may prevent future complications for both the pregnant women and the neonate. The publications cited in the article are up-to-date, consistent with the latest scientific reports (excluding one citation: Wong, S. F., Chow, K. M., Leung, T. N., Ng, W. F., Ng, T. K., Shek, C. C., Ng, P. C., Lam, P. W. Y., Ho, L. C.,To, W. W. K., Lai, S. T., Yan, W. W., & Tan, P. Y. H. (2004). Pregnancy and perinatal outcomes of women with severe acute respiratory syndrome. American Journal of Obstetrics and Gynecology, 191(1), 292–297.) An article uses scientific, evidence-based methods to provide up-todate data to improve the quality of healthcare. The statistical analysis was described in detail and carried out in a way that allowed to obtain statistically significant results. The results are clearly presented, the tables are properly described, which enables their analysis and interpretation. The subject of the article corresponds to the magazine recipients interests, moreover, as already mentioned, it is a global topic that arouses global interest not only among gynecologists and obstetricians, but also other specialists who work at the health of women and children. The level of language is properly adapted to the recipients, understandable. The included keywords correspond to the topic of the article
Author Response
The reviewed article: „SARS-CoV-2 infection in pregnancy: clues and proofs of adverse outcomes” is related to the problem that has taken over the world - the SARS-CoV-2 pandemic. This publication presents the results of research related to: SARS-CoV-2 virus infection in pregnant women, vertical transmission of the virus from mother to neonate, complications related to the neonates born to SARS CoV-2 infected mothers. The authors of the article properly defined the research problem, which allowed to obtain reliable results and identified the association between SARS-CoV2 infection, obstetric outcomes and vertical transmission. The research methodology is accurate and the study group properly selected (large study and control group) according to the purpose of the study. Patients were included in the study in accordance with the protocol approved by ethical committee of the General Hospital 84 Policlinico Umberto I, University of Rome “Sapienza” (Ref. 6275). Inclusion and exclusion criteria are apropriately selected for the subject of the study - patients from both (the control and the study) group participating in the study signed consents and confirmed that they had not been vaccinated against SARS-CoV-2 infection. Performed examinations were carried out in accordance with the established diagnostic methods. Biological materials were properly stored and secured, samples taken to identify infection were collected and stored in appropriate conditions in accordance with WHO guidelines and evaluated by specialized diagnostic laboratories. Keeping in mind that SARS-CoV-2 is global-wide infection and the population of women participating in the study was identified, this research might encourage other research groups to expand the following subject by including populations in other parts of the world.
Thank you for the punctual summary of our work.
The conclusions in the article are consistent with the results contained in the table (Table 5. Characteristics of symptomatic SARS-CoV-2 positive mothers and their children as compared to healthy controls) however, in order to properly assess their pathophysiology, each topic should be widened.
We changed the conclusions as follows:
“Pregnant women affected by SARS-CoV-2 are more likely to develop se-vere obstetric outcomes and their newborns are more prone to have a low birth weight and low arterial pH, due to the clear signs of malperfusion and chronic intervillositis. Despite SARS-CoV-2 vertical transmission seems a ra-re event, probably related to the potential viral entry into the bowel lumen through the fetal swallowing of infected amniotic fluid, it can still occur, although the pathophysiological aspects need further investigations.”
Never the less this article might be an introduction to further challenges and scientific researches on the correlation of SARS-CoV-2 infection with complications in pregnant women and neonates. According to the paragraphs 4 (Discussion) and 5 (Conclusions) contained in the article and the conclusions detailed by the researchers, COVID-19 vaccines are recoemnded for pregnant women and may prevent future complications for both the pregnant women and the neonate.
Thank you for enhancing one of the strengths of our work. We absolutely support the importance of vaccination during pregnancy, for both mothers and newborns. Moreover, we would have liked to compare our data with those obtained from vaccinated patients, but the SARS-CoV-2 vaccination was not available in Italy when we designed the study and recruited patients. Therefore, we added in the text (page 19 line 390): “In the future, it will be interesting to investigate the effect of SARS-CoV-2 vaccination on adverse obstetric outcomes, since SARS-CoV-2 vaccination was not recommended in Italy for pregnant women when we recruited the patients for this study.”
The publications cited in the article are up-to-date, consistent with the latest scientific reports (excluding one citation: Wong, S. F., Chow, K. M., Leung, T. N., Ng, W. F., Ng, T. K., Shek, C. C., Ng, P. C., Lam, P. W. Y., Ho, L. C.,To, W. W. K., Lai, S. T., Yan, W. W., & Tan, P. Y. H. (2004). Pregnancy and perinatal outcomes of women with severe acute respiratory syndrome. American Journal of Obstetrics and Gynecology, 191(1), 292–297.)
Thank you. We replaced the reference with more recent evidence: “Metz TD, Clifton RG, Hughes BL, Sandoval GJ, Grobman WA, Saade GR, Manuck TA, Longo M, Sowles A, Clark K, Simhan HN, Rouse DJ, Mendez-Figueroa H, Gyamfi-Bannerman C, Bailit JL, Costantine MM, Sehdev HM, Tita ATN, Macones GA; National Institute of Child Health and Human Development Maternal-Fetal Medicine Units (MFMU) Network. Association of SARS-CoV-2 Infection With Serious Maternal Morbidity and Mortality From Obstetric Complications. JAMA. 2022 Feb 22;327(8):748-759. doi: 10.1001/jama.2022.1190. PMID: 35129581; PMCID: PMC8822445.”
An article uses scientific, evidence-based methods to provide up-todate data to improve the quality of healthcare. The statistical analysis was described in detail and carried out in a way that allowed to obtain statistically significant results. The results are clearly presented, the tables are properly described, which enables their analysis and interpretation. The subject of the article corresponds to the magazine recipients interests, moreover, as already mentioned, it is a global topic that arouses global interest not only among gynecologists and obstetricians, but also other specialists who work at the health of women and children. The level of language is properly adapted to the recipients, understandable. The included keywords correspond to the topic of the article.
Thank you. We sincerely trust to carry significant support for maternal counselling in case of SARS-CoV-2 infection during pregnancy.